# Pattern of Antibiotic Use in the Perinatal Period in a Public University Hospital in Romania

**DOI:** 10.3390/medicina58060772

**Published:** 2022-06-07

**Authors:** Viviana Hodoșan, Cristian Marius Daina, Dana Carmen Zaha, Petru Cotrău, Adriana Vladu, Carmen Pantiș, Florica Ramona Dorobanțu, Marcel Negrău, Adriana Maghiar, Lucia Georgeta Daina

**Affiliations:** 1Faculty of Medicine and Pharmacy, Doctoral School of Biomedical Sciences, University of Oradea, 1 University Street, 410087 Oradea, Romania; pinta.vivi@yahoo.com (V.H.); petrucotrau@yahoo.com (P.C.); 2Psycho-Neurosciences and Recovery Department, Faculty of Medicine and Pharmacy, University of Oradea, 1 University Street, 410087 Oradea, Romania; lucidaina@gmail.com; 3Department of Preclinical Disciplines, Faculty of Medicine and Pharmacy, University of Oradea, 1 University Street, 410087 Oradea, Romania; dorobanturamona@yahoo.com (F.R.D.); maghiar_adi@yahoo.com (A.M.); 4Department of Surgical Disciplines, Faculty of Medicine and Pharmacy, University of Oradea, 1 University Street, 410087 Oradea, Romania; adrianavladu68@yahoo.com (A.V.); pantisc@yahoo.com (C.P.); negrau.marcel@gmail.com (M.N.)

**Keywords:** pregnancy, labor, antibiotics, fetal, risk

## Abstract

*Background and Objectives*: Antibiotics are the most frequently prescribed drugs in hospitals and their prescription is increased during pregnancy and labor. There are limited data about this issue, and the safe use of antibiotics in pregnancy and antibiotic resistance remains a concern. The aim of the study is to evaluate the use of antibiotics among pregnant women attending hospital for five years. *Materials and Methods*: Antibiotic consumption and treatment information of patients were retrospectively collected from a hospital software program and expressed as defined daily dose (DDD) according to the World Health Organization (WHO) methodology for inpatients between 2017 and 2021. We evaluated antibiotic prescription by name, classes, and Food and Drug Administration (FDA) categories. *Results*: Antibiotic consumption shows a decreasing trend between 2017 and 2019, but an increasing one between 2020 and 2021. Ceftriaxone was the most prescribed antibiotic in each year, followed by cefixime, amoxicillin, metronidazole, cefuroxime, ampicillin, and ciprofloxacin. We noticed that first- and fourth-generation cephalosporins were not prescribed to these patients. A very small percentage of women in this study received antibiotics such as aminoglycosides; fluoroquinolones were generally contraindicated in the perinatal period. A large percentage of prescriptions were antibiotics classified as category B by the FDA. The most common infections that occurred in the perinatal period were those of the kidney and urinary tract in a higher number than in other studies. *Conclusions*: Our study shows that many classes of antibiotics used in perinatal women belong to category B antibiotics, the most prescribed being cephalosporins. Because of insufficient safety evidence and the potential for teratogenic effects on the fetus, restricted use among the category C and D antibiotic classes was seen and anticipated. Improving maternal health requires the involvement of healthcare experts in risk assessment and evaluation of existing data for appropriate antibiotic selection, dose, duration of medication, and monitoring.

## 1. Introduction

Antibiotic safety is essential when considering its use during the perinatal period. Antibiotics can have short- or long-term effects on the fetus [1]. The degree of teratogenicity depends on the gestational time of the pregnancy, the dose and duration of therapy, and the degree of drug transfer across the placenta. In addition, there are important genetic and environmental factors involved.

Although pregnancy is a physiological condition, hormone fluctuations and changes in immunity are responsible for the predisposition to infection during pregnancy. The most frequent antibiotic prescriptions during pregnancy are for the treatment of urinary and respiratory tract infections [2]. Infections that occur during pregnancy can affect the mother and baby and are considered risk factors for health complications. It is estimated that antibiotics are among the most prescribed medications during pregnancy for urinary infections. However, urinary tract infections (UTIs) are the most common diseases in the population and during pregnancy contributing to the increase in antibiotic use [3]. UTIs in the perinatal period, including asymptomatic bacteriuria, are associated with exposure of the fetus to antibiotics, at the same time with maternal morbidity and adverse pregnancy outcomes. Some infections that develop during pregnancy can be followed by miscarriage, preterm labor, or birth issues [4].

The most common diagnostics for prescription of antibiotics in the perinatal period are chorioamnionitis, genital, and urinary infections, but there are some prophylactic purposes for their administration such as prevention of neonatal sepsis by group B streptococcus (GBS) and prophylaxis in cesarean deliveries. Antibiotics are prescribed also to prevent preterm labor, but there is no evidence supporting this in the absence of signs of infections [5]. Current clinical practices highlight the identification of women with GBS colonization to prevent the transmission of bacteria to the newborn infant [6].

There are data about increasing antibiotic prescriptions in pregnancy, and antibiotics represent a higher percentage of all medication prescribed in pregnancy: one in four pregnant women is prescribed an antimicrobial [7,8]. Irrational antibiotic use is significant in perinatal medicine for numerous reasons. Pregnant women and their fetuses/infants are in their most vulnerable time in gestation and neonatal period, and during this time and the first year after delivery antibiotic abuse is at its highest level. Antibiotic overuse is linked to the emergence of antimicrobial resistance (AMR) as well as to changes in the immune system. The intestinal microbiota plays a crucial role in the immune system’s development and external stimuli can easily disturb the establishment of the intestinal microbiota [9]. Disruption of the intestinal microbiota during this sensitive stage could have a significant impact on immunological development. Numerous studies have linked the makeup of the gut microbiota to a variety of immune and non-immune disorders, including sepsis [10] and neonatal necrotizing enterocolitis, and it has a potential role in the development of chronic inflammatory bowel disease, diabetes mellitus, and allergy illnesses in children [11,12,13].

There is insufficient evidence about the safety of antibiotics administration during pregnancy because of many reasons. First, clinical trials in pregnant women are limited, and the teratogenicity of certain drugs is unknown [14]. Second, there are not so many reports. Only around 10% of antibiotics used during pregnancy are supported by safety data, according to a review [15]. The FDA in the United States amended and issued the Pregnancy and Lactation Labeling Rule in 2014, a useful tool for other countries also [16], and antibiotics were classified into ‘A to X’ safety categories.

There is currently no standardized way of using antibiotics in pregnant women that is sensible, safe, and successful. The availability of clinical evidence in this population is limited due to ethical constraints. Prescribers treat pregnant women based on the trimester of pregnancy, the severity of the disease, and potential fetal harm using a risk–benefit assessment. Antibiotic use during pregnancy is being impacted by the rising challenge of antibiotic resistance, with overuse and misuse of these medications being the primary causes of AMR. The AMR epidemic is changing how antibiotics are used, increasing death and morbidity while also driving up healthcare costs. Antibiotic management and use have clinical, economic, and environmental consequences [17].

The objective of this study was to evaluate the use of antibiotics and trends profile among pregnant women attending a public hospital, as well as conditions treated. Evaluation can indicate the measures that are required in order to maintain the safety of antibiotic treatments for both newborns and mothers.

## 2. Materials and Methods

This is a retrospective study conducted for five years from 2017 to 2021 in the County Clinical Emergency Hospital of Oradea, Romania. Data about pregnant inpatients treated with J01 class antibiotics were collected from the hospital software program InfoWorld which electronically stores patient files and treatment information. Details about antibiotic use and diagnosis were analyzed. Antibiotic name, dose, route of administration, and diagnosis were recorded. Individual patients’ written informed consent for the data collection was not needed, but it was obtained at admission. Antibiotic prescription pattern was expressed in grams and defined daily dose (DDD) per 100 bed days according to the WHO Collaborating Centre for Drug Statistics Methodology, ATC/DDD Index 2022 [18]. Hospital consumption of antimicrobials was evaluated by calculation of the total number of grams of each antimicrobial used during the period of interest and the result was divided by the WHO-assigned DDD resulting in the number of DDD (DDDs). Antimicrobial use density was expressed as DDDs/100 bed days, for each antibiotic, class, route of administration, and risk categories described by FDA, A, B, C, D, and X as can be seen in the Table 1 [19].

Descriptive statistics were performed using Excel. The Mann–Kendall test was used to express (upward or downward) trends. A null hypothesis was considered if there was no trend in the series and an alternative hypothesis if there was a trend. The null hypothesis cannot be rejected if the estimated *p*-value is greater than the significance level of 0.05. Ethical approval was obtained from the Ethics Committee of the County Clinical Emergency Hospital of Oradea.

## 3. Results

The antibiotics prescribed to the patients enrolled in our study as well as their amounts converted to DDD during the evaluated period can be seen in Table 2. Antibiotic consumption shows a decreasing trend between 2017 and 2019, but with an increasing one in 2020–2021. Most prescribed antibiotics were ceftriaxone, cefixime, amoxicillin, metronidazole, cefuroxime, ampicillin, ciprofloxacin, cefaclor, gentamicin, clindamycin, amoxicillin/clavulanate, representing 98.17%.

Less prescribed were oxacillin, azithromycin, ceftazidime, vancomycin, benzyl penicillin, cefoperazone/sulbactam, colistin, imipenem/cilastatin, moxifloxacin, and cefoperazone. The first ten prescribed antibiotics are relatively the same in the five years evaluated. Ceftriaxone was the most prescribed followed by cefuroxime between 2017 and 2020, while cefixime and ceftriaxone were the most prescribed in 2021. Among the ten most prescribed antibiotics, amoxicillin/clavulanate was replaced by clarithromycin in 2019, ciprofloxacin in 2020, and meropenem in 2021 (Figure 1). More than 86% of the antibiotics were administrated by the oral route.

Assessment of antibiotic use by class shows a high consumption of third-generation cephalosporins, followed by aminopenicillins, imidazoles, second-generation cephalosporins, fluoroquinolones, aminoglycosides, lincosamide, macrolides, and carbapenems. Less prescribed were penicillins (natural and izoxazolilpenicillins), glycopeptides, polypeptides, and rifampicins for the same five years (Table 3). About half or more of cephalosporins third-generation, aminopenicillin, imidazoles, fluoroquinolones, aminoglycosides, and lincosamides were prescribed in 2021 at the same time with a decrease in the administration of second-generation cephalosporins. Third-generation cephalosporins and imidazoles were prescribed in close quantities between 2017 and 2020, but with a significant increase in 2021. The same aspect, i.e., a marked increase in the prescription in 2021, also presented fluoroquinolones, aminoglycosides, lincosamides, macrolides, and carbapenems. The prescription of second-generation cephalosporins and natural penicillins, on the other hand, showed a pronounced decreasing trend or have not been prescribed at all (polypeptides, rifampicins).

A large percentage (92.34%) of prescriptions were antibiotics classified as category B by the FDA with the remaining 7.66% being categories C and D. As can be seen in Table 4, the consumption of FDA category B antibiotics remains almost constant between 2018 and 2020 with a decrease in 2021, while the consumption of category C antibiotics shows an increasing trend. Although overall category D antibiotics showed low values for prescriptions and included only amikacin, the downward trend between 2018 and 2020 is followed by a significant increase in 2021.

Cephalosporins (third and second generation), aminopenicillins, and imidazoles were the most frequent class of FDA category B antibiotics prescribed to pregnant women followed by lincosamides, carbapenems, and macrolides. Natural penicillins, ceftazidime, cefoperazone, cefoperazone/sulbactam, azithromycin, and glycopeptides were slightly less prescribed from the same FDA category B antibiotics. No first and fourth-generation cephalosporins were prescribed (Table 5).

Table 6 shows the prescribed category C antibiotics, containing those drugs for which animal studies may or may not have been undertaken to reveal adverse effects, but for which there are no suitable and well-controlled trials in pregnant women. Gentamycin, fluoroquinolones, and clarithromycin were the most prescribed category C drug classes. Amikacin (FDA antibiotic category D) was used less than gentamicin (FDA antibiotic category B).

The incidence of infections during the evaluated period was 25.86% (4090 infections reported in 17,146 births). Table 7 shows the main categories of infections and their incidence.

It is noted that the main infections encountered are those of the urinary tract including asymptomatic and symptomatic bacteriuria; the values are higher than those reported by international studies. There is also a decrease in the incidence of this status in 2020 compared to 2017–2019, possibly by limiting the possibilities of travel and performing consultations and excessive medical maneuvers, due to pandemic restrictions. In our study, the incidence of chorioamnionitis shows an almost constant trend except for 2018, when it registered a decrease; but overall, during the four years evaluated, it remained low (1.89%). Surgical wound infections, and mastitis had a low incidence in the analyzed period.

## 4. Discussion

The gestational period is an important factor that complicates antibiotic therapy. Multiple studies advise against the use of drugs in the first trimester as the fetus is still developing [8]. An analysis of antibiotic usage in the second and third trimesters performed by Thinkhamrop et al. found no difference in the likelihood of congenital abnormalities between the two trimesters, but the same authors concluded that there was insufficient evidence to fully evaluate possible fetal harm [20].

The literature indicates that beta-lactams remain widely prescribed in pregnancy representing approximately 65% of all antibiotics, with penicillins accounting for 30% [8]. Similar results were obtained from our study, indicating that 81.4% of all antibiotics prescribed to pregnant women belonged to a beta-lactam class. The available data on the use of cephalosporins during pregnancy do not suggest an increased risk of spontaneous miscarriage or congenital abnormalities at therapeutic doses, and few studies identified a possible association with cardiovascular defects in exposed offspring, low birth weight, and preterm delivery [21]. A higher percentage (64.14%) was identified in women being treated with cephalosporins and aminopenicillins (16.77%) during pregnancy; consistent with the current trends of antibiotic therapy in pregnancy [22]. Pregnancy increased the levels of estrogens and deposition of glycogen in the vaginal microbiota including the community of microorganisms that reside in the lower reproductive tract. Pregnancy increases *Candida* colonization which causes vaginal infections and could be involved in spontaneous preterm birth [23,24,25]. Imidazole (only metronidazole in our study), cephalosporins, and aminopenicillins are the most used antibiotics in pregnancy in our study. Although metronidazole is classified as pregnancy category B and is guideline-recommended therapy for bacterial vaginosis and *Trichomonas* infections in pregnancy, it should be avoided in the first trimester of pregnancy [26,27].

The prescribed antibiotics were classified by class of active substance and FDA safety categories B, C, and D. As expected, more than 92.34% of pregnant women received category B of antibiotics according to the FDA because there is no evidence to demonstrate a risk to the fetus. The studies show a long record of safety data and represent 17.3% of all prescriptions expressed in DDD/100 bed days during the five years evaluated. We noticed that the most used were amoxicillin followed by ampicillin, while amoxicillin/clavulanate was only 5.87% of prescriptions expressed in DDD/100 bed days with a decreasing trend; therefore, aminopenicillins are the second more prescribed in our hospital. Aminoglycosides cross the placenta and should be administrated if the benefits outweigh the risks. Between 2018 and 2020, aminoglycosides followed a downward trend, with an increasing prescription in 2021 consumption. Amikacin (FDA antibiotic category D) was used less than gentamicin. Quinolones are antibiotics that are routinely used to treat a wide range of infections. There are reports showing that quinolones are not associated with unfavorable pregnancy outcomes, but their safety profile in pregnant women remains debatable [28].

Yefet et al. conducted a systematic evaluation of the safety of quinolones in pregnancy and found no serious congenital abnormalities. Although fluoroquinolones have been proven to have potentially hazardous effects in several animal models, the research suggests that it is usually at higher dosages. The FDA, on the other hand, amended its cautions for fluoroquinolones, both oral and injectable, in 2016 [29]. The authors showed that when the medicine is taken systemically, it might cause severe and perhaps irreversible side effects on the fetus, such as disruption of tendons, joints, muscles, and nerves. It may also cause type 2 diabetes, in addition to the previously listed consequences. Quinolones are classified as FDA category C, with significant restrictions to their use. Ciprofloxacin, ofloxacin, and moxifloxacin together accounted for 5.18% of all antibiotics expressed in DDD/100 bed days during the five years evaluated. The most prescribed were ciprofloxacin and ofloxacin, and less prescribed was moxifloxacin.

Clindamycin is classified as pregnancy category B, and it was the only lincosamide antibiotic used for pregnant women in our hospital. Clindamycin prescription displayed a decreasing trend during the first four years evaluated, but with a significant increase in 2021.

Macrolides are pregnancy category A prescribed in 0.55% of all antibiotics expressed in DDD/100 bed days during the five years evaluated. Prescribing macrolide during pregnancy is not uncommon as similar results have been reported elsewhere [30,31]. The use of macrolides (category B) in pregnancy is, however, a growing concern. Significantly, a recent study by Fan et al. followed 104605 children from birth to 14 years of age and it was concluded that prescribing macrolides in any trimester was associated with an increased risk of genital malformation, whereas a previous cohort of 1033 women exposed to macrolides (erythromycin, azithromycin, clarithromycin, or roxithromycin) reported that there was no association with this drug and the development of major abnormalities in the fetus [32].

Carbapenems are extended-spectrum beta-lactams and are typically reserved for severe infections and/or in case of pathogens resistant to penicillins and cephalosporins. Meropenem and imipenem/cilastatin (category C) were administrated to patients enrolled in this study. The use of these antibiotics during pregnancy was minimal, but it displays a significant and worrying increasing trend in 2021, in contrast to the colistin which belongs to the same category of reserve antibiotics as FDA category C. Pregnancy causes many changes to a woman’s body that result in an increased risk of developing urinary tract infections. *E. coli* is the common cause of urinary tract infections followed by *Enterococcus faecalis*, *Staphylococcus saprophyticus*, *Proteus mirabilis*, *Klebsiella pneumoniae*, and *Chlamydia*. The difference between pregnant and non-pregnant women is the increase in the incidence of asymptomatic bacteriuria in pregnant women. The incidence of urinary tract infections in pregnant women is 0.3–1.3%, similar to that in non-pregnant women, but recurrent bacteriuria is more common in pregnant women than non-pregnant women [32]. A study in Israel of 200,000 pregnant women showed a 2.5% incidence of asymptomatic bacteriuria and 2.3% of symptomatic urinary tract infections. In this population, bacteriuria was associated with complications: hypertension, diabetes, intrauterine growth retardation, and premature birth [33]. In our study, the incidence of asymptomatic and symptomatic bacteria in pregnant women who gave birth at our hospital was higher compared to the average observed from international studies [34,35]. Urinary tract infections (UTIs) during pregnancy are the most common causes of antibiotic prescriptions. Even in asymptomatic bacteriuria, patients are treated with antibiotics because they can lead to complications such as cystitis, pyelonephritis, and preterm birth [35,36]. In our study, nitrofurantoin (category B), an antibiotic specific for urinary tract infections, was not prescribed. Some studies reported nitrofurantoin might increase the risk of hemolytic anemia in pregnant patients with severe glucose-6-phosphate dehydrogenase deficiency [37], an association between nitrofurantoin use during pregnancy, and cleft lip and palate, while other studies did not demonstrate any link between nitrofurantoin exposure in women and major congenital malformation [14]. For these reasons, nitrofurantoin use as an option for the treatment of urinary tract infections in pregnant women should be reconsidered, and the hospital has evaluation and decision-making tools, respectively [38,39]. In recent years, the epidemiology of many diseases has changed both with regard to the host as well as to the bacteria’s spectrum and susceptibility to antibiotics [40]. Chorioamnionitis is an ascending infectious and inflammatory process of the membrane that surrounds the fetus due to bacterial, fungal, or viral agents originating from the genitourinary tract and is associated with a risk of preterm labor, retinopathy of prematurity, bronchopulmonary dysplasia, neonatal sepsis, or neonatal death [41,42,43]. Chorioamnionitis can also occur hematogenously by maternal bacteremia or by contamination of the amniotic cavity because of an invasive procedure (amniocentesis). A 2014 study described an incidence of 9.7 per 1000 births [44]. In developed countries (Canada, Western Europe, Australia), the incidence is equal to or slightly lower compared to the United States. In poorly developed countries, premature rupture of membranes is closely correlated with chorioamnionitis, premature births, and increased mortality. The most common antibiotics recommended in the primary management of chorioamnionitis were ampicillin and gentamicin, and alternative antibiotics were clindamycin [45]. During the five years evaluated, the number of chorioamnionitis remains low (1.89%) compared with other results from other studies showing an almost double incidence [46]. Postpartum infections are pathological entities that occur after a vaginal birth or by cesarean section or during breastfeeding. Many studies show an incidence of postpartum infectious complications ranging from 5.5 to 7.4%. The range for endometritis is 1.6 (0.9–2.5), for surgical wound infection 1.2% (1.0–1.5), and for maternal peripartum infection 1.1 (0.3–2.4) [46]. It should be noted that in our study, only the first three (wounds, genitourinary infections, and mastitis) were the most common postpartum complications noted and their incidence is similar to the data reported by Woodd et al. Our study has a few limitations. It was conducted in one hospital and included only inpatients for five years. Neither the duration of therapy nor the exposure to adverse effects of therapy and recurrences was assessed. There was insufficient follow-up on those patients to identify whether their children were affected by the antibiotic therapy received. Despite these limitations, this study contributes to knowledge in this special patient group and builds on the evidence that currently exists. Long-term reliance on antibiotics considered safe to use could be followed by the development of resistance, requiring the use of broad-spectrum or alternative medicines whose safety has already been completely established [47].

## 5. Conclusions

There is currently a relative scarcity of data regarding antibiotic use in pregnancy and our research should improve the knowledge in this field. Our study shows that many classes of antibiotics used in perinatal women belong to category B antibiotics, the most prescribed being cephalosporins. Because of insufficient safety evidence and the potential for teratogenic effects on the fetus, restricted use among the category C and D antibiotic classes was seen and anticipated. A very small percentage of women in this study received antibiotics such as aminoglycosides, and fluoroquinolones were generally contraindicated in the perinatal period unless the benefit of the drug exceeded the risk. Improving maternal and child health requires the involvement of healthcare experts in risk assessment and evaluation of existing data for appropriate antibiotic selection, dose, duration of medication, and monitoring. It is also essential that more studies follow up on the side effects that drugs have on pregnant women.

## Figures and Tables

**Figure 1 medicina-58-00772-f001:**
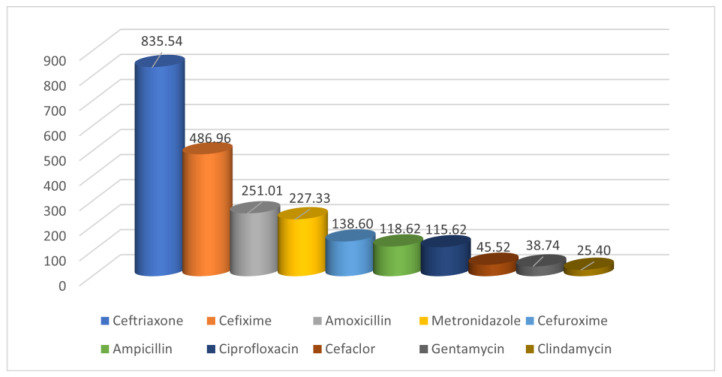
The first ten antibiotics prescribed from 2017–2021.

**Table 1 medicina-58-00772-t001:** Antibiotic FDA categories.

Category	Evidence	Classes or Antibiotics
A	There is no evidence of risk to the fetus in the first and later trimesters.	
B	In case of animals and pregnant women there is no evidence to demonstrate risk to the fetus.	Penicillin and beta-lactam inhibitors, carbapenems (except imipenem/cilastatin), glycopeptides, imidazole, lincosamides, macrolides, third-generation cephalosporins
C	Although there is evidence of detrimental effects on the fetus in animals and no adequate and well-controlled studies in humans, potential advantages may justify the drug’s usage in pregnant women despite potential dangers.	Fluoroquinolones, polymixyns, rifamycins, trimethoprim-sulfamethoxazole, gentamycin
D	Based on adverse reaction data from investigational or marketing experience or human trials, there is positive evidence of human fetal risk, although possible benefits may justify use of the medicine in pregnant women despite potential hazards.	Aminoglycosides—except gentamycin Tetracyclines
X	There has been evidence of fetal abnormalities in animals or humans, and/or there is positive evidence of human fetal risk based on adverse reaction data from experimental or marketing experience, and the dangers of using the medicine in pregnant women clearly outweigh the possible benefits.	

**Table 2 medicina-58-00772-t002:** Prescribed antibiotics expressed in DDD/100 bed days during the five years evaluated.

ATC	Antibiotic	DDD/100 Bed Days					
		2017	2018	2019	2020	2021	*p*-Value
J01GB06	Amikacin	0.61	0.37	0.13	0.10	8.55	1
J01GB03	Gentamicin	7.41	2.56	1.80	2.21	24.74	0.73
J01CA04	Amoxicillin	24.71	33.04	15.88	31.48	145.88	0.73
J01CA01	Ampicillin	28.77	15.17	4.06	7.83	62.78	0.73
J01CR02	Amoxicillin/clavulanate	15.36	1.55	1.15	0.84	5.87	1
J01DH02	Meropenem	0.55	1.00	0.44	0.83	6.34	0.33
J01DH51	Imipenem/cilastatin	-	-	-	0.01	0.13	0.14
J01DC04	Cefaclor	19.72	18.51	4.32	2.95	0.00	0.08
J01DC02	Cefuroxime	40.42	47.28	27.49	21.41	1.97	0.08
J01DD04	Ceftriaxone	129.72	117.06	113.90	127.98	346.86	0.30
J01DD12	Cefoperazone	0.01	-	0.001	-	0.00	-
J01DD02	Ceftazidime	0.53	0.24	0.10	0.33	0.11	0.53
J01DD08	Cefixime	33.06	16.93	12.63	22.69	401.63	0.30
J01DD62	Cefoperazone/ sulbactam	0.02	0.01	0.008	-	0.12	1
J01XD01	Metronidazol	13.58	12.34	15.80	9.39	176.21	0.73
J01MA02	Ciprofloxacin	2.89	1.17	0.49	2.36	108.69	0.33
J01MA01	Ofloxacin	0.80	0.33	0.32	1.16	3.54	0.30
J01MA14	Moxifloxacin	0.03	-	0.032	-	0.00	1
J01CF04	Oxacillin	0.88	0.34	0.04	0.03	0.23	0.73
J01FF01	Clindamycin	6.45	2.97	2.27	1.66	12.04	1
J01FA09	Clarithromycin	1.02	1.16	2.45	1.02	2.00	1
J01FA10	Azithromycin	0.33	0.18	0.006	0.078	0.79	0.73
J01FA01	Erythromycin	-	-	0.35	1.058	2.66	0.08
J01CE01	Benzyl penicillin	0.14	0.08	0.104	0.069	0.04	0.30
J01XA01	Vancomycin	0.03	0.10	0.12	0.040	0.58	0.73
J01XB01	Colistin	0.11	0.039	-	-	-	0.37
Total		327.30	272.54	204.05	235.61	1311.77	

**Table 3 medicina-58-00772-t003:** Prescribed antibiotics by class expressed in DDD/100 bed days during the five years evaluated.

Antibiotics by Class	2017	2018	2019	2020	2021	Total	*p*-Value
Cephalosporins (third generation)	163.37	134.26	126.65	151.01	748.73	1324.02	1
Aminopenicillins	68.85	49.77	21.10	40.17	214.53	394.41	0.73
Imidazoles	13.59	12.34	15.80	9.39	176.21	227.33	0.72
Cephalosporins (second generation)	60.15	65.80	31.82	24.37	1.97	184.12	0.08
Fluoroquinolones	3.74	1.51	0.86	3.53	112.22	121.86	0.30
Aminoglycosides	8.03	2.94	1.95	2.32	33.29	48.53	0.73
Lincosamide	6.45	2.97	2.27	1.67	12.04	25.40	1
Macrolides	1.36	1.35	2.82	2.17	5.46	13.16	0.30
Carbapenems	0.56	1.01	0.45	0.84	6.47	9.33	0.73
Izoxazolilpenicillins (oxacillin)	0.89	0.35	0.05	0.03	0.23	1.55	0.73
Glycopeptides	0.03	0.11	0.12	0.04	0.58	0.88	0.73
Natural penicillins	0.14	0.08	0.10	0.07	0.04	0.44	0.30
Polypeptides	0.11	0.04	-	-	-	0.15	0.37
Rifampicins	0.03	0.00	0.06	-	-	0.09	1

**Table 4 medicina-58-00772-t004:** Prescribed antibiotics by FDA category expressed in DDDs/100 bed days.

Category of Antibiotics	2017	2018	2019	2020	2021	Total	*p*-Value
B	313.48 (95.77)	266.53 (97.79)	198.69 (97.37)	228.69 (97.06)	1163.88 (88.82)	2171.27 (92.34)	0.73
C	4.88 (1.49)	2.71 (0.99)	3.31 (1.62)	4.57 (1.93)	139.09 (10.60)	168.57 (7.16)	0.30
D	0.61 (0.18)	0.37 (0.13)	0.13 (0.06)	0.10 (0.04)	8.55 (0.65)	9.79 (0.41)	1

**Table 5 medicina-58-00772-t005:** FDA category B antibiotics commonly prescribed expressed in DDDs/100 bed days.

ATC	Antibiotic	2017	2018	2019	2020	2021	Total
J01DD04	Ceftriaxone	129.73	117.06	113.90	127.98	346.86	835.54
J01DD08	Cefixime	33.07	16.94	12.63	22.69	401.63	486.96
J01CA04	Amoxicillin	24.72	33.04	15.88	31.49	145.88	251.01
J01XD01	Metronidazole	13.59	12.34	15.80	9.39	176.21	227.33
J01DC02	Cefuroxime	40.42	47.29	27,50	21.42	1.97	138.60
J01CA01	Ampicillin	28.77	15.17	4,06	7.84	62.78	118.62
J01DC04	Cefaclor	19.73	18.52	4.32	2.95	-	45.52
J01FF01	Clindamycin	6.45	2.97	2.27	1.67	12.04	25.40
J01CR02	Amoxicillin/ clavulanate	15.36	1.55	1.16	0.85	5.87	24.78
J01DH02	Meropenem	0.56	1.01	0.45	0.83	6.34	9.18
J01FA01	Erythromycin	-	-	0.36	1.06	2.66	4.07
J01FA10	Azithromycin	0.34	0.19	0.01	0.08	0.79	1.41
J01DD02	Ceftazidime	0.54	0.25	0.11	0.34	0.11	1.35
J01XA01	Vancomycin	0.03	0.11	0.12	0,04	0.58	0.88
J01CE01	Benzyl penicillin	0.14	0,08	0.10	0,07	0.04	0.44
J01DD62	Cefoperazone/ sulbactam	0.03	0.01	0.01	0.00	0.12	0.17
J01DD12	Cefoperazone	0.01	-	-	-	-	0.01

**Table 6 medicina-58-00772-t006:** FDA category C antibiotics commonly prescribed expressed in DDDs/100 bed days.

ATC	Antibiotic	2017	2018	2019	2020	2021	Total
J01GB03	Gentamycin	7.41	2.56	1.81	2.22	24.74	38.74
J01DH51	Imipenem/ cilastatin	-	-	-	0.01	0.13	0.14
J01MA02	Ciprofloxacin	2.90	1.17	0.50	2.37	108.69	115.62
J01MA01	Ofloxacin	0.81	0.34	0.32	1.17	3.54	6.17
J01MA14	Moxifloxacin	0.04	-	0.03	-	-	0.07
J01FA09	Clarithromycin	1.02	1.16	2.46	1.03	2.00	7.68
J01XB01	Colistin	0.11	0.04	-	-	-	0.15

**Table 7 medicina-58-00772-t007:** Number of patients and infections occurred in between 2017 and 2021.

Diagnostic	2017	2018	2019	2020	2021	Total	*p*-Value
			Number (%)			
Renal and urinary tract infections	813(22.38)	1050(29.47)	604 (18.78)	241(7.27)	470(13.72)	3178(18.53)	0.30
Chorioamnionitis	69(1.89)	48(1.34)	74(2.30)	69(2.08)	87(2.54)	347(2.02)	0.30
Surgical wound infections of obstetric origin	0(0)	4(0.11)	4(0.12)	7(0.21)	36(1.05)	51(0.29)	0.14
Other genito-urinary tract infections after birth	76(2.09)	97(2.72)	128(3.98)	111(3.35)	78 (2.27)	490(2.85)	0.73
Mastitis	2(0.05)	4(0.11)	1(0.03)	10(0.30)	7(0.20)	24(0.13)	0.73
Total number of births	3632	3562	3216	3311	3425	17,146

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
