# Peer review of "Pattern of Antibiotic Use in the Perinatal Period in a Public University Hospital in Romania"

_medicina, 2022, doi:10.3390/medicina58060772_

Round 1

Reviewer 1 Report

The authors have studied the Pattern of antibiotic use during perinatal period in a public 2 university hospital in Romania. Please see my suggestions regarding this manuscriptȘ

  1. FDa must be explained before abbreviating it. According to the Instructions for Authors, Acronyms/Abbreviations/Initialisms have been defined the first time they appear in each of three sections: the abstract; the main text; under the first figure or table. When defined for the first time, the acronym/abbreviation/initialism should be added in parentheses after the written-out form”. Please check and revise the entire manuscript in this regard.
  2. L109-111. Detail more the aim of the study. What novelty your paper brings to the field? What are the special aspects that differentiate your paper from others in the same topic you can mention?
  3. Table 7, 2nd Merge all rows (from 2nd to 7th) and write just once Number (%). The Table will be cleaner and not so repetitive.
  4. Remove that sentence. The aim of the study was already mentioned at the final of Introduction section. It is repetitive.
  5. Remove the empty spaces between the paragraphs, in the same section. The text in each section must be compact (please see the Instructions for authors in this regard), and the aspect of the manuscript more professional.
  6. 40 has nothing to do with the topic. I suggest replacing it with Zaha D.C., et al.https://doi.org/10.1016/j.scitotenv.2019.06.076

Author Response

Dear reviewer

On behalf of my coauthors, I would like to thank you for the opportunity to revise and resubmit our manuscript. We found the reviewers’ comments to be helpful in revising the manuscript and have carefully considered and responded to each suggestion. The revisions are marked up using the “Track Changes” function as you suggested.

Thank you again for your consideration of our revised manuscript.

Sincerely,

Dana Carmen Zaha

Rev 1

The authors have studied the Pattern of antibiotic use during perinatal period in a public 2 university hospital in Romania. Please see my suggestions regarding this manuscript.

FDa must be explained before abbreviating it. According to the Instructions for Authors, “Acronyms/Abbreviations/Initialisms have been defined the first time they appear in each of three sections: the abstract; the main text; under the first figure or table. When defined for the first time, the acronym/abbreviation/initialism should be added in parentheses after the written-out form”. Please check and revise the entire manuscript in this regard. -solved

L109-111. Detail more the aim of the study. What novelty your paper brings to the field? What are the special aspects that differentiate your paper from others in the same topic you can mention? -solved

Table 7, 2nd Merge all rows (from 2nd to 7th) and write just once Number (%). The Table will be cleaner and not so repetitive. -solved

Remove that sentence. The aim of the study was already mentioned at the final of Introduction section. It is repetitive. -solved

Remove the empty spaces between the paragraphs, in the same section. The text in each section must be compact (please see the Instructions for authors in this regard), and the aspect of the manuscript more professional.-we corrected

40 has nothing to do with the topic. I suggest replacing it with Zaha D.C., et al. https://doi.org/10.1016/j.scitotenv.2019.06.076-solved

Reviewer 2 Report

The authors tried to address an important problem in the care of pregnant women.  However, there are many areas for improvement in this paper.

There are many grammatical errors that need to be corrected and the abstract and the paper checked for English sentence structure.

Please do not use abbreviations without first explaining, such as DDD use in abstract.

Methods are not adequately described in the abstract.

Introduction and Discussion:  too long with many of the sections redundant and repeating.  Introduction can be cut in half.

Reference 18 in methods does not make sense.

Results:

tables are partially redundant and mostly can be changed to "supplemental tables".

No statistical analysis was performed.  You state that there is a trend up or down, but it is impossible to state whether that is a significant trend or not.

Discussion: very drawn out and with redundancies.

Author Response

Dear reviewer

On behalf of my coauthors, I would like to thank you for the opportunity to revise and resubmit our manuscript. We found the reviewers’ comments to be helpful in revising the manuscript and have carefully considered and responded to each suggestion. The revisions are marked up using the “Track Changes” function as you suggested.

Thank you again for your consideration of our revised manuscript.

Sincerely,

Dana Carmen Zaha

The authors tried to address an important problem in the care of pregnant women.  However, there are many areas for improvement in this paper.

There are many grammatical errors that need to be corrected and the abstract and the paper checked for English sentence structure. -solved

Please do not use abbreviations without first explaining, such as DDD use in abstract. -solved

Methods are not adequately described in the abstract. -solved

Introduction and Discussion:  too long with many of the sections redundant and repeating.  Introduction can be cut in half. -solved

Reference 18 in methods does not make sense. -solved

Results:

tables are partially redundant and mostly can be changed to "supplemental tables".-we could consider them as supplemental files, but they support results and could be helpful to follow them

No statistical analysis was performed.  You state that there is a trend up or down, but it is impossible to state whether that is a significant trend or not. -solved, we add a statistical test, Mann-Kendall test.

Discussion: very drawn out and with redundancies. -solved.

Round 2

Reviewer 2 Report

Thank you for working on the revisions and making some changes to the manuscript.

You were able to correct some of my previous concerns, however, there are still grammatical errors.  The sentence in the methods of the abstract that was added is now a fragment.

Methods of the manuscript can still be improved by being more specific in regards to the typical use of antibiotics in your medical system.

Discussion still needs to be decreased as it is overly lengthy and redundant.

Author Response

Thanks for your valuable suggestions on our article.
We corrected, hopefully, the mistakes of expression and grammar.
We also shortened the discussions
